# Molecular Epidemiology of *Pseudomonas aeruginosa* in Brazil: A Systematic Review and Meta-Analysis

**DOI:** 10.3390/antibiotics13100983

**Published:** 2024-10-17

**Authors:** Yan Corrêa Rodrigues, Marcos Jessé Abrahão Silva, Herald Souza dos Reis, Pabllo Antonny Silva dos Santos, Daniele Melo Sardinha, Maria Isabel Montoril Gouveia, Carolynne Silva dos Santos, Davi Josué Marcon, Caio Augusto Martins Aires, Cintya de Oliveira Souza, Ana Judith Pires Garcia Quaresma, Luana Nepomuceno Gondim Costa Lima, Danielle Murici Brasiliense, Karla Valéria Batista Lima

**Affiliations:** 1Bacteriology and Mycology Section, Evandro Chagas Institute (SABMI/IEC), Ministry of Health, Ananindeua 67030-000, PA, Brazil; herald.reis@live.com (H.S.d.R.); antonnypabllo@gmail.com (P.A.S.d.S.); danielle-vianna20@hotmail.com (D.M.S.); isabelmontoril13@gmail.com (M.I.M.G.); carollyne83@hotmail.com (C.S.d.S.); davijosuemarcon@gmail.com (D.J.M.); cintyaoliveira@iec.gov.br (C.d.O.S.); anaquaresma@iec.gov.br (A.J.P.G.Q.); luanalima@iec.gov.br (L.N.G.C.L.); daniellemurici@iec.gov.br (D.M.B.); 2Program in Epidemiology and Health Surveillance (PPGEVS), Evandro Chagas Institute (IEC), Ministry of Health, Ananindeua 67030-000, PA, Brazil; 3Program in Parasitic Biology in the Amazon Region (PPGBPA), State University of Pará (UEPA), Tv. Perebebuí, 2623-Marco, Belém 66087-662, PA, Brazil; 4Department of Health Sciences (DCS), Federal Rural University of the Semi-Arid Region (UFERSA), Av. Francisco Mota, 572-Bairro Costa e Silva, Mossoró 59625-900, RN, Brazil; caio.aires@ufersa.edu.br

**Keywords:** *Pseudomonas aeruginosa*, high-risk clones, MLST, antimicrobial resistance, molecular epidemiology, regional variability, Brazil

## Abstract

Background: Globally, *Pseudomonas aeruginosa* is a high-priority opportunistic pathogen which displays several intrinsic and acquired antimicrobial resistance (AMR) mechanisms, leading to challenging treatments and mortality of patients. Moreover, its wide virulence arsenal, particularly the type III secretion system (T3SS) *exoU^+^* virulotype, plays a crucial role in pathogenicity and poor outcome of infections. In depth insights into the molecular epidemiology of *P. aeruginosa*, especially the prevalence of high-risk clones (HRCs), are crucial for the comprehension of virulence and AMR features and their dissemination among distinct strains. This study aims to evaluate the prevalence and distribution of HRCs and non-HRCs among Brazilian isolates of *P. aeruginosa*. Methods: A systematic review and meta-analysis were conducted on studies published between 2011 and 2023, focusing on the prevalence of *P. aeruginosa* clones determined by multilocus sequence typing (MLST) in Brazil. Data were extracted from retrospective cross-sectional and case-control studies, encompassing clinical and non-clinical samples. The analysis included calculating the prevalence rates of various sequence types (STs) and assessing the regional variability in the distribution of HRCs and non-HRCs. Results: A total of 872 samples were analyzed within all studies, of which 298 (34.17%) were MLST typed, identifying 78 unique STs. HRCs accounted for 48.90% of the MLST-typed isolates, with ST277 being the most prevalent (100/298—33.55%), followed by ST244 (29/298—9.73%), ST235 (13/298—4.36%), ST111 (2/298—0.67%), and ST357 (2/298—0.67%). Significant regional variability was observed, with the Southeast region showing a high prevalence of ST277, while the North region shows a high prevalence of MLST-typed samples and HRCs. Conclusions: Finally, this systematic review and meta-analysis highlight the role of *P. aeruginosa* clones in critical issue of AMR in *P. aeruginosa* in Brazil and the need of integration of comprehensive data from individual studies.

## 1. Introduction

Due to the globally increasing threat of antimicrobial resistance (AMR) in the human, animal, and environmental interfaces, the World Health Organization (WHO) pooled several bacterial pathogens urgently needing epidemiology studies, enhanced surveillance, and novel antimicrobial drugs discovery and development, in which *Pseudomonas aeruginosa* appears as a ‘high-priority’ pathogen, mainly due to the dissemination of carbapenem-resistant strains (CR) exhibiting difficult-to-treat resistance (DTR), extensively drug-resistance (XDR), and even pan-drug-resistance (PDR) phenotypes [1,2,3,4,5].

In Brazil, the overall resistance rates of clinically relevant bacterial species, including *P. aeruginosa*, indicate that the AMR situation is similar to European countries, such as Turkey and Greece, but opposite to that reported in Spain and Sweden [6]. In four countries in the Latin America—Bolivia, Brazil, Paraguay, and Peru—a significant increasing trend in non-susceptibility to carbapenems was identified, while in these other three countries—Argentina, Guatemala, and Panama—decreasing trends were noticed since 2014 [7]. As a reflection of this scenario, national surveillance data on AMR recently reported *P. aeruginosa* as the third mostly isolated bacterial pathogen in Brazilian intensive care units (ICUs) in 2021, after the peak of the COVID-19 pandemic period. Indeed, during the pandemic, *P. aeruginosa* was among the most frequently causative agents of ventilator-associated pneumonia (VAP) [8,9]. CR and MDR *P. aeruginosa* isolates were responsible for up to 34.4% of primary bloodstream infections (BSI) and up to 43.0% of urinary tract infections (UTI). Additionally, as a troubling consequence, the increased use of last-line antimicrobials, such as polymyxin and colistin, in clinical practice has led to resistance rates as high as 25.0% among these isolates [10].

Among the several intrinsic and acquired AMR-mediating mechanisms of *P. aeruginosa*, (I) the activity/overexpression of efflux pumps of the resistance-nodulation-division (RND) family, mainly the MexAB-OprM efflux pump; (II) chromosomally encoded AmpC cephalosporinases overexpression; (III) intrinsic loss or decrease in the porin protein OprD, and (IV) acquisition of carbapenemases, especially metallo-β-lactamases, are the most commonly described among CR and MDR/XDR strains [11,12,13]. Moreover, the secretion of multiple virulence factors by *P. aeruginosa* strains plays a critical role in bacterial survival and host damage, and often negatively impact hospitalized patients’ prognosis [14,15]. Considered the most clinically relevant and extensively characterized virulence determinant, the type III secretion system (T3SS), particularly cytotoxic *exoU^+^* strains, have been linked to fluoroquinolones and carbapenems resistance, polymicrobial infections, mortality of hospitalized patients, and intermittent infection and early clinical alterations among cystic fibrosis (CF) patients [16,17,18,19,20,21,22].

Along with the emergence and detection of highly resistant strains combining distinct resistance and virulence features, it has been crucial to apply typing methods for analysis of clonal relationships, epidemiological surveillance, population structure studies, outbreaks identification, and environmental reservoirs of *P. aeruginosa* strains [23,24,25]. The multilocus sequence typing (MLST) scheme for *P. aeruginosa* genotyping developed by Curran et al. [26] remains as the reference/gold standard method for the identification of clones based on the sequence type (ST) definition with more than 5000 STs described so far (Accessed on 10 August 2024) [27].

Single and unrelated STs, with local endemicity and importance, mainly compose the polyclonal population structure of observed *P. aeruginosa* strains. Nevertheless, widespread clones selected at high antibiotic pressure settings (e.g., healthcare institutions) are referred to as international and epidemic high-risk clones (HRCs), which are responsible for most of the MDR/XDR strains prevalent in hospital-acquired infections (HAIs) cases, mortality of patients, and restrictions on antimicrobial therapy regimens. Importantly, the *P. aeruginosa* HRCs ST235, ST111, ST233, ST244, ST357, ST308, ST175, ST277, ST654, and ST298 have been demonstrated as a major concern [28,29,30,31]. Indeed, several molecular epidemiology-based studies evaluating clinical and non-clinical Brazilian samples demonstrated the circulation of different HRCs; however, the HRC ST277 has been described as endemic and responsible to the remarkable spread of the *bla*_SPM-1_ (São Paulo Metallo-β-lactamase-1) in Brazil [29,32,33].

Considering the importance of comprehensive data on the molecular epidemiology based on MLST, as well as the surveillance of HRCs and their role in the dissemination of AMR and virulence mechanisms on human, animal, and environment settings, the present study aims to provide a condensed summary on the genetic diversity and prevalence of *P. aeruginosa* clones reported by MLST genotyping in Brazilian territory.

## 2. Material and Methods

### 2.1. Study Design

This work consists of a systematic review and meta-analysis to assemble succinct and up to date information on the molecular epidemiology based on the MLST of *P. aeruginosa* strains reported in Brazil. The present study followed the recommendations of the Preferred Report Items of a Systematic Review and Meta-Analyses (PRISMA) Statement and was registered with the International Prospective Register of Systematic Reviews (PROSPERO) under registration number CRD42024588801 [34].

A ratio-type meta-analysis was performed in this study for the correlated investigations: (I) to verify the prevalence of MLST genotypes among *P. aeruginosa* isolates in Brazil, (II) to determine and compare the prevalence of HRCs and non-HRCs, and (III) to determine the prevalence of the ST277 in comparison to other HRCs within the analyzed studies. The POT strategy was used, and it consisted of the following question: “What is the prevalence of MLST genotypes (STs) among *P. aeruginosa* isolates in Brazil?” The anagram for its formation was composed of “P” for problem (*P. aeruginosa* STs), “O” for outcome (frequency/prevalence of *P. aeruginosa* STs in Brazil), and “T” for study type (original studies). The search terms that were used for the search in the title, abstract, and keywords fields based on medical subject headings (MeSH) on the PubMed database were as follows: “Humans” OR “Animals” AND “*Pseudomonas aeruginosa*/classification” OR “*Pseudomonas aeruginosa*/genetics” OR “*Pseudomonas aeruginosa*/pathogenicity” AND “Brazil”. On LILACS and SciELO databases, the health sciences descriptors (DeCS) applied were the following: “*Pseudomonas aeruginosa*” AND “Brazil”. Additionally, reference lists of the selected articles were reviewed to verify further relevant articles.

### 2.2. Eligibility Criteria and Definitions

Cross-sectional studies, case-control studies, and cohort studies were all considered as having available, full open access in Portuguese, English, or Spanish between 2010 and July 2024 on PubMed, LILACS, and SciELO. This time frame was used to condense updates from the previous 14 years. Articles published before 2010, short reports (communications), genome notes, duplicates, abstract-only availability, inaccessibility to key information in the article, and subjects unrelated to the research query were among the exclusion criteria. This led to the creation of the end sample, which reflected all phases, inclusions, and exclusions.

Also, studies included in the present meta-analysis consisted of reports assessing the genetic diversity of *P. aeruginosa* isolates from human, animal, and environment settings, describing the presence of at least two isolates genotyped by MLST (with defined ST results). Reported STs and allelic profiles from each study were double checked to match the available data at the PubMLST database (http://pubmlst.org/paeruginosa, accessed on 10 August 2024). PHYLOViZ 2.0 and Bionumerics v6.6 platforms were used for data management and the analysis of clonal complexes (CCs), which were defined by related ST clusters exhibiting variation in a single locus (single locus variants—SLV), and the construction of the minimum spanning tree (MST) was based on the goeBURST Full algorithm [35]. Finally, HRCs and emerging HRCs were considered according to those previously listed [29,31].

### 2.3. Data Extraction

Data collection, extraction, and analysis were performed independently by two researchers (HSR and PASS), which organized the data in tabular form with the aid of Microsoft Office Excel 365. In case of disagreement, discussions were resolved by two other researchers (YCR and MJAS). All data were collected and reviewed in June and July 2024 and included the following: (1) title; (2) year; (3) database; (4) type of study; (5) sample/isolates period; (6) region; (7) setting; (8) number of samples/isolates; (9) number of MLST-genotyped samples/isolates; (10) MLST genotypes; (11) resistance phenotype; (12) results of resistance markers; and (13) results of virulence markers.

### 2.4. Methodological Quality and Risk of Bias Assessment

For the analysis of methodological quality, the 13 included papers were evaluated using Joanna Briggs Institute (JBI) checklists, consisting of a series of questions organized according to the study design. Each checklist is specific to a type of research, allowing reviewers to determine whether the methods used were appropriate and whether the results are reliable. The answers to these questions can be “Yes”, “No”, “Uncertain”, or “Not Applicable” which helps in assessing the robustness of the reviewed studies [36].

The accuracy of the scores was only considered if the answers were “Yes” to the questions on the items in each checklist. In this sense, two checklists were used in this review, the JBI Checklist for Analytical Cross-Sectional Studies (scores ranging from 0 to 8) and the JBI Checklist for Case-Control Studies (scores ranging from 0 to 10) [37]. If the score of each article scored more than or equal to 60% of the total points, it was approved for inclusion in this item [38].

To evaluate the risk of bias in non-randomized controlled studies, the Non-Randomized Studies of Interventions (ROBINS-I) instrument was used [39]. The recovered articles were subjected to both independent risk of bias assessments and methodological evaluation by YCR and HSR. Any disagreement was settled through discussion by a third author (MJAS).

### 2.5. Statistical Analysis

Using the fixed effects model, the summary event rate and associated 95% confidence intervals (CIs) were calculated. Subgroup analyses were performed based on the geographical regions of Brazil. Heterogeneity between studies was assessed using the I^2^ statistic and the chi-square test, with a significance threshold of *p* ≤ 0.05. A meta-regression was performed to determine the predictors of the HRC status, with the year of publication and number of samples included as predictor variables. Potential publication bias was evaluated using funnel plots and Begg’s test. Funnel plots were constructed to visualize the distribution of effect sizes (prevalence rates) against their standard errors. The Comprehensive Meta-Analyses—CMA program, version 2.2 (Biostat, Englewood, NJ, USA) was used on a computer to perform the statistical analyzes of this meta-analysis.

## 3. Results

### 3.1. Literature Search

By applying the selection criteria, a total of 425 articles were retrieved from the PUBMED, SciELO, and LILACS databases on the topic for reading. However, 392 articles were letters to the editor, duplicates, short communications, studies unavailable in full version, outside the intended languages, or contained information not relevant to the research question, and were therefore excluded. By applying the eligibility criteria, 33 studies were read in full, independently and in pairs, which resulted in the final sample of the 13 studies included (Figure 1).

### 3.2. Characterization of Included Studies and Molecular Epidemiology Data

A total of 13 studies were included in the final meta-analysis, all of which were presented in English, published between 2011–2023, and sourced from PubMED and LILACS databases. These studies were conducted by authors affiliated with Brazilian teaching and research institutions: nine (69.22%) from the Southeast region, two (15.42%) from the North, and two (15.42%) from the Northeast regions. Most of the studies (92.31%) were retrospective cross-sectional studies, with one retrospective case-control study (7.69%), including samples from clinical/hospital settings, as well as environmental (non-clinical) samples (water and animal sources) obtained between 1994–2021 (Table 1).

A total of 872 *P. aeruginosa* isolates were evaluated within studies, with only 298/872 (34.17%) genotyped by the MLST, revealing 78 unique STs (Figure 2). Specifically, 48.90% (146/298) of the isolates were identified as HRCs, including ST277 (100/298—33.55%), ST244 (29/298—9.73%), ST235 (13—4.36%), ST111 (2/298—0.67%), and ST357 (2/298—0.67%). In addition, HRCs were detected over different time periods in Brazil since 1994, as presented in Figure 3. The remaining isolates (152/298—51.10%) were related to non-HRCs, distributed among 73 different STs, of which ST804 (11/298—3.69%), ST1859 (8/298—2.68%), ST2524 (8/298—2.68%), ST1602 (5/298—1.67%), ST1860 (5/298—1.67%), ST2236 (5/298—1.67%), and ST2237 (5/298—1.67%) were the most detected (Table 1, Figure 2 and Figure 3). Suggested emerging HRCs were identified, ST245, ST253, ST446, and ST532, but for analysis were considered as non-HRCs (Figure 2, Figure 3 and Figure 4). Furthermore, clonal complex (CC) relationships were evidenced, including the following: CC244 (ST244, ST595, and ST594), CC235 (ST235 and ST593), CC1560 (ST1560, ST1944, and ST2236), CC498 (ST498 and ST252), CC1602 (ST1602 and ST1419), CC1603 (ST1603 and ST446), and CC1767 (ST1767 and ST1768) (Figure 2).

All studies included CR isolates, predominantly associated with MDR and XDR phenotypes. The *bla*_SPM-1_ gene emerged as the most frequently detected resistance marker, followed by *bla*_KPC_, aminoglycoside-modifying enzymes (AMEs), and efflux pump overexpression. Regarding virulence, only six studies investigated virulence-related genes, with a particular focus on the T3SS, where notably, the cytotoxic (*exoU^+^*) and invasive (*exoS^+^/exoU^−^*) virulotypes were identified in 41 and 117 isolates, respectively (Appendix A). Particularly, several HRCs and non-HRCs were detected harboring MβL genes in Brazil (Figure 5).

The methodological quality assessment revealed a high level of quality, and the risk of bias analysis showed a low risk of bias among studies for the included studies (Figure 6).

### 3.3. Results and Publication Bias of Meta-Analysis of Proportion of P. aeruginosa HRCs and Non-HRCs Strains

A total of 13 studies were evaluated to determine the prevalence rate for HRCs and non-HRCs among the MLST-genotyped isolates, revealing the combined prevalence rate of 41.20% (0.412) (95% CI = 0.347–0.481) (Figure 7). The statistical analysis showed significant differences among the studies (χ^2^ = 74.41; df = 12; *p* < 0.001; I^2^ = 83.87%), indicating high heterogeneity. A subgroup group analysis was performed based on Brazilian geographic regions: In the North region, the prevalence rate of MLST-genotyped samples was 46.20% (0.462) (95% CI = 0.337–0.592), but analysis also showed significant variability among the studies within this region (χ^2^ = 9.41; df = 1; *p*-value = 0.002; I^2^ = 89.37%). The Northeast region had a prevalence rate of 43.90% (95% CI = 0.248–0.649), and it was observed as a low level of heterogeneity with no statistical significance (χ^2^ = 1.90; df = 1; *p*-value = 0.168; I^2^ = 47.43%). In the Southeast region, the prevalence rate was much lower at 38.50% (95% CI = 0.305 to 0.473), and with a mean difference in heterogeneity among the studies within this region (χ^2^ = 62.09; df = 8; *p*-value < 0.001; I^2^ = 87.11%) (Figure 7).

To evaluate the potential publication bias in the meta-analysis of the proportion of HRCs and non-HRCs, a funnel plot was constructed, and Begg’s test was conducted. The funnel plot displays the distribution of effect sizes (prevalence rates) against their standard errors (Figure 8). The results of the Begg’s test revealed a significant correlation with a Kendall’s tau coefficient of 0.23 and a *p*-value of 0.27, indicating no publication bias.

As high heterogeneity was found among the studies investigated in this first analysis, a meta-regression was conducted in terms of the year of publication of each study in fixed model effects, indicating a statistical significance and suggesting the potential bias source (χ^2^ = 56.01; df = 8; *p*-value < 0.001) (Figure 9).

### 3.4. Results and Publication Bias of Meta-Analysis of ST277 within the HRCs Genotypes

A proportion analysis was also performed from the perspective of the MLST-genotyped data from the samples from these studies. In this sense, the prevalence of ST277 in relation to other HRC STs found in these studies was determined. Only one study of the 13 evaluated in the previous analysis was now ineligible as no HRCs were detected (Silva et al. [43]). The analysis revealed a prevalence rate of 54% (95% CI = 0.412–0.663). In relation to geographic subgroups, the North region presented a prevalence of 42.8% (95% CI = 0.242–0.636). The Northeast had a low frequency rate of 14% (95% CI = 0.02–0.47). Regarding the Southeast region, it presented the highest rate of 69% or 0.690 (95% CI = 0.523–0.818) (Figure 10). Regarding the heterogeneity between studies, the Northeast region was the only one that did not present a significant difference (χ^2^ = 0.101; df = 1; *p*-value = 0.75; I^2^ = 0%), while the other analyses for both subgroups and overall revealed high heterogeneity in the data (overall [χ^2^ = 46.175; df = 11; *p*-value < 0.001; I^2^ = 0%]; North [χ^2^ = 11.665; df = 1; *p*-value < 0.001; I^2^ = 91.42%]; Southeast [χ^2^ = 25.171; df = 7; *p*-value < 0.001; I^2^ = 72.19%]) (Figure 10).

A funnel plot was created to assess the publication bias in the meta-analysis of the MLST genotypes (Figure 11). The funnel plot illustrates the distribution of prevalence rates against their standard errors. The Begg’s test was performed to quantitatively assess publication bias, yielding a Kendall’s tau coefficient of 0.03 and a *p*-value of 0.89. These results indicate no significant correlation, suggesting that there is no evidence of publication bias in this meta-analysis.

As high heterogeneity was found among the studies investigated, a meta-regression was performed in terms of the sample size of each study in fixed model effects, indicating a statistical significance (a positive association between sample size and HRC classification) and suggesting the potential bias source (coefficients = −1.0983; 95% CI = −2.02–−0.16; *p*-value = 0.0014) (Figure 12).

## 4. Discussion

This systematic review and meta-analysis provide a comprehensive overview on the molecular epidemiology of *P. aeruginosa* in Brazil, emphasizing the prevalence and distribution of HRCs and non-HRCs identified through MLST genotyping. In this study, 78 unique STs were identified among 298 isolates, reflecting considerable genetic diversity (Figure 2). Our findings also reveal a significant prevalence and diversity of HRCs (ST111, ST244, ST235, ST277, and ST357), along with the remarkable dominance of ST277. Non-HRCs made up 51.10% of the MLST-typed isolates, underscoring the presence of several STs with potential epidemiological significance. Moreover, strains were observed in clonal expansion, as six CCs were detected including at least two distinct STs (CC244, CC235, CC498, CC1560, CC1602, CC1603, and CC1767). This genetic prospect is in sense with data from various studies on *P. aeruginosa* molecular epidemiology, which consistently show extensive clonal diversity and most isolates being represented by unique and distinct genotypes, reflecting a non-clonal epidemic population structure [25,32].

ST277 is a particularly problematic *P. aeruginosa* HRC recognized for its extensive distribution and critical impact on the AMR scenario in Brazil, heightening public health concerns nationwide. Since its first detection in an oncology patient in São Paulo (Southeast region), this clone has played a major role in the dissemination of the *bla*_SPM-1_ gene across various clinical settings in the country [33,40,46]. Indeed, there are a few reports of ST277 outside of Brazil, as it has been documented in Japan and China related to IMP-producing strains, and in the UK, related to an XDR (colistin-susceptible only) isolate [53,54,55]. Even though ST277 is endemic to Brazil, 87 isolates deposited in the PubMLST database belonging to this clone (last accessed 10 August 2024) have been reported in Austria, Australia, Brazil, Central African Republic, China, France, and Ghana [27]. Oppositely, this dominance is well documented in the Brazilian territory, with a prevalence rate of 54% (95% CI = 0.412–0.663) and reports of its presence in both clinical and environmental samples in states from different regions, including São Paulo, Rio de Janeiro, Pernambuco, Acre, and Pará states. Such a high prevalence further highlights the clone’s adaptability and ability to thrive in various environments subjected to intense antibiotic pressure, such as hospitals, where it exacerbates the difficulty of managing infections caused by the MDR/XDR *P. aeruginosa*.

Our data showed that ST277 was the dominant HRC in the Southeast region, identified in up to 89.0% of isolates in some studies, particularly in clinical settings, as deeply explored by Silva et al. [40], which was one of the first studies providing MLST and other genotyping (PFGE and ribotyping) data for the SPM-1-producing *P. aeruginosa* in Brazil, and De Oliveira Santos et al. [48], which documented a significant presence of ST277 in clinical isolates from Rio de Janeiro over a 21-year period. Alarmingly, only in the Southeast region, studies evaluating non-clinical isolates (environmental samples) were conducted; Esposito et al. [50] raises concerns about impacted urban river environments potentially serving as reservoirs and sources of re-infection, especially in densely populated urban areas with significant hospital infrastructure; likewise, Martins et al. [46] identified migratory birds in Brazil harboring the ST277 strains in their microbiota, implying that these birds may have played a crucial role in the broad dissemination of this clone within the territory, along with the *bla*_SPM-1_ gene. This presence indicates a potential cycle of transmission between hospitals and the surrounding environment, further complicating efforts to control its spread. In the North region, ST277 among clinical isolates was identified in two studies conducted by our research group; Rodrigues et al. [49], which firstly provided data on MLST genotyping in the region, and Dos Santos et al. [52], describing a possible outbreak of ST277 in healthcare institutions in the North region. In the Northeast, ST277 presence was reported by Cavalcanti et al. [41] and related to one CR-PA isolate. This data scenario reflects the clone’s substantial, yet variable, impact across the country.

Curiously, *P. aeruginosa* ST277 exhibits a unique and highly conserved set of AMR and virulence genes. Key resistance markers commonly identified in ST277 strains include *aac(6′)-Ib’*, *aadA7*, *aph(3′)-IIb*, *bla*_OXA-56_, *bla*_PDC-374_, *bla*_SPM-1_, *catB7*, *cmx*, *crpP*, *fosA*, and *rmtD1*, while for virulence, *aprA*, *lasA*, *lasB*, *toxA*, *exo S*, *exoT*, *and exoY* genes are identified, as well as the O2 serotype group. These genes are largely chromosomally encoded, contributing to the stability and persistence of this clone in nosocomial infections [56,57]. The integration of these AMR genes into genomic islands like PAGI-15 and PAGI-25 further underscores the adaptive success of ST277 within Brazilian healthcare settings, where it poses a significant challenge due to its extensive drug resistance. In addition, the resistome and virulome of ST277 is not only extensive but also largely restricted to Brazilian strains, highlighting a strong geographical and evolutionary link. The study by Dos Santos et al. [52] recently revealed that ST277 containing this resistome and virulome likely played a crucial role in its persistence and spread during the COVID-19 pandemic in the North region This has further complicated treatment options and strongly corroborates previous findings regarding the clone’s impact on AMR in Brazil [50,56,58].

CC/ST244 is a globally distributed *P. aeruginosa* HRC that is frequently associated with MDR and even non-MDR infections, suggesting a prevalence not fully related to AMR [30,31,32,59]. Its presence has been recorded across various regions in Asia, Europe, North America, and Oceania, mainly due to its involvement in hospital outbreaks and association with multiple AMR genes, including those conferring resistance to aminoglycosides, carbapenems, and macrolides, as well as *exoS^+^* and O2 serogroup virulence features [27,60,61,62]. This HRC has been widely reported in China, where it has been implicated in IMP- and VIM-producing isolates causing CR *P. aeruginosa* infections among burn and ICU patients [30,62,63,64,65]. Furthermore, in-depth genetic analysis of ST244 strains from this country demonstrated a significant relation to those from North America, suggesting either a shared evolutionary ancestor or frequent strain exchanges between these regions [63]. In Spain, the ST244 isolates were found in multiple regions, indicating interregional dissemination, particularly associated with the production of VIM-2 carbapenemase, and even co-producing both VIM-2 and KPC-2 carbapenemases, further complicating treatment options due to the dual resistance mechanisms [66]. ST244 is not limited to clinical environments; it has also been detected in various other sources across different continents. Likewise in Spain and China, this clone has been detected on children’s fecal samples and on animal-derived foods, indicating the adaptability of ST244, its potential role as a community reservoir, and its ability to colonize diverse ecological niches, which could contribute to its spread within both healthcare and community settings [67,68].

In Brazil, CC/ST244 has been identified as the second most prevalent significant *P. aeruginosa* HRC, particularly in the Southeast and North regions. According to the data presented in the systematic review, ST244 has been detected in both clinical and environmental samples among 29 (9.73%) of the MLST-typed isolates in seven studies, demonstrating its widespread presence. Data from Miranda et al. [42] show that in the Southeast region of Brazil (RJ state), this clone was mainly detected in non-clinical settings, particularly in hospital wastewater treatment plants, even across various stages of the treatment process. Such isolates exhibited MDR/XDR and non-resistant phenotypes, associated with critical resistance genes such as *bla*_VIM_ and *bla*_KPC_. In the North region, our research group reported ST244 was the most prevalent HRC among the tested strains from 2010–2013, being predominantly found in adult ICUs and harboring the *bla*_CTX-M_ (five isolates) and *exoU^+^* genes (two isolates) [49]. In the Northeast region, this clone was found only among clinical samples and related to the spread of the *bla*_KPC_ gene and harbors in the CRISPR–Cas adaptive immune systems [41,51]. Finally, this data draw attention to the significant role of ST244 in the dissemination of *bla*_VIM_ and *bla*_KPC_ variants in Brazil, mirroring its impact on a global scale [31,32].

CC/ST235 is recognized as the most remarkable and widely disseminated *P. aeruginosa* HRC worldwide contributing to the prevalence of XDR/DTR strains in healthcare settings across all continents, mainly in Europe and Asia [38,59,69,70,71,72]. ST235 has remarkable genomic adaptability, which includes a heightened ability to acquire and integrate foreign resistance elements through homologous recombination, that further distinguishes it from other HRCs. The DprA determinant, specifically identified in ST235, enhances this adaptability, contributing to the clone’s resilience and ability to persist in diverse environments [31,32]. This success of ST235 is also closely linked to its association with over 60 β-lactamase variants, worryingly those strains harboring *bla*_KPC_, *bla*_GES_, *bla*_FIM_, *bla*_IMP_, *bla*_VIM,_ and *bla*_NDM_ variants, which further amplifies the clinical challenges associated with this clone [29,30,73]. Beyond its resistance features, ST235 is characterized by enhanced virulence due to the presence of the ExoU cytotoxin delivered via the T3SS and O11 serotype, key factors driving the severe clinical outcomes associated with ST235 infections [16,17,28]. Interestingly, in non-human and non-clinical samples, *P. aeruginosa* ST235 has been recently linked to infection among companion animals (dogs and cats) in Japan and Thailand, and in VIM-producing isolates recovered from hospital wastewater in Brazil [71,74,75].

Across various regions of Brazil, ST235 was identified in four studies on clinical samples, accounting for 4.36% (13/298) of the MLST-typed *P. aeruginosa* isolates. This presence is notable nationwide, even though it is less dominant compared to other HRCs like ST277 and ST244. In the Southeast region, Brüggemann et al. [47] conducted a genomic analysis of 20 *P. aeruginosa* strains isolated over two decades in São Paulo. Among these, three strains were identified as ST235, which clustered closely with other Brazilian and Argentinean isolates, suggesting regional relatedness. These ST235 strains carried multiple resistance determinants, including *aac(6′)-Ib-cr*, which confers resistance to both aminoglycosides and fluoroquinolones, and the *exoU* gene. In Rio de Janeiro, Cacci et al. [44] described this clone as related to both MDR and non-resistant isolates, but did not clearly define AMR markers related to this clone. In the Northeast region, Cavalcanti et al. [41] identified ST235 among clinical isolates collected in Pernambuco and highlighted two *P. aeruginosa* isolates of this clone co-harboring both *bla*_GES_ and *bla*_KPC_, which might have posed a significant treatment challenge in the region. Rodrigues et al. [49] identified ST235 in five of MLST-typed *P. aeruginosa* isolates from a referral hospital in Pará, highlighting its presence in the North region of Brazil since 2010. All those ST235 isolates were classified as MDR/XDR; three of these isolates were clonally related and dispersed across different ICUs harboring the *bla*_CTX-M-1_ gene, and four of these also presented the cytotoxic virulotype (*exoU^+^*). Collectively, these factors indicate the critical importance of *P. aeruginosa* ST235 as a relevant lineage in the national and global context of AMR, also necessitating ongoing surveillance and strategic infection control measures.

The HRCs ST111 and ST357 highlight their significant role in the adaptability and persistence of *P. aeruginosa* isolates across healthcare settings. Both clones have been detected in diverse regions, showing a strong presence in Europe, Asia, Africa, and South America (Poland, Singapore, Malaysia, Sweden, Croatia, Austria, Spain, Norway, France Hungary, Senegal, Nigeria, Egypt, Ghana, and Brazil) [76,77,78]. The resistance profiles of ST111 and ST357 pose substantial challenges to clinical treatment. ST111 is strongly associated with the production of VIM-2 in Europe, along with other β-lactamases such as GES and PSE, which confer resistance to a broad spectrum of β-lactam antibiotics [13,30,79,80,81,82]. Similarly, ST357 has usually been linked to IMP-producing isolates, with recent reports of NDM-producing isolates, with both mechanisms contributing to resistance to carbapenems [80,83,84,85,86,87]. As for virulence, ST111 typically lacks the ExoU cytotoxin, exhibiting considerable virulence through the *exoS^+^* invasive virulotype and is often associated with the O12 serotype. In contrast, *P. aeruginosa* ST357 strains usually produce the potent ExoU cytotoxin and are related to the O11 serotype, similarly to ST235 [28,31,32].

Both ST111 and ST357, even though identified in smaller proportions according to this study (4/298—1.34%), represent significant concerns due to their high resistance and virulence profiles. Recent findings from [49,51] emphasize their restricted spread in the Brazilian territory and the importance of surveillance for such clones, as both are significant contributors to HAIs. ST111 was found only in the State of Pará (North region) harboring the *bla*_CTX-M_ gene in the adult ICU at a referral hospital, while in the State of Pernambuco (Northeast region), ST357 was suggested as an emerging threat in the clinical environment of ICUs, especially considering its association with the CRISPR/Cas systems, which could confer advantages in resisting bacteriophage attacks and maintaining resistance genes.

According to del Barrio-Tofiño et al. [30], other STs have been suggested as emerging HRCs of *P. aeruginosa*, which have been detected in at least three countries and are often associated with MDR/XDR phenotypes. These clones frequently harbor intrinsic horizontally acquired β-lactamases, contributing to their enhanced resistance profiles and making them a significant concern for global public health. Among them, ST245, ST253, ST446, and ST532 were detected among Brazilian isolates in the Southeast and Northeast regions. STs 245 and 253 were related to one isolate each, detected co-harboring the *bla*_OXA-50_ and *bla*_PAO_ genes, and were associated to invasive and cytotoxic virulotypes, respectively [47]. A genome note also described the presence of a polymyxin-resistant isolate assigned to ST245 causing VAP in São Paulo [88]. In the city of Recife, ST446 was related to one isolate presenting an overexpression of the MexAB-OprM system, and in Rio de Janeiro, ST532 was detected on a susceptible clinical isolate [41,44]. Globally, ST245 has been reported in China among hospital inpatients and in fecal samples from healthy individuals in Spain [63,89]. The PA14 strain (ST253) has been widely detected in several countries associated with the *exoU^+^* virulotype [90,91]. In the United Kingdon (UK) and Germany, ST466 was associated with a SIM-producing isolate and was colonizing patients at a pulmonary clinic [92,93]. Finally, ST532 was related to an isolate harboring a ciprofloxacin-modifying enzyme (CrpP) in South Africa and among patients with bronchiectasis in Spain [94,95].

Non-HRCs constituted 51.10% of the MLST-typed isolates, with 73 different STs identified. Among these, ST804, ST1602, ST1859, ST1860, ST2524, ST2236, and ST2237 were the most frequently detected. Although these STs are not classified as HRCs, they still play a major role in the genetic diversity and epidemiology of *P. aeruginosa* as their presence in distinct environments may contribute to the dissemination of resistance and virulence factors. Interestingly, most of the predominant non-HRCs (ST1602, ST1859, ST1860, ST2236, and ST2237) were firstly detected in Rio de Janeiro, Southeast Brazil. De Oliveira Santos [48] is one of the few studies reporting the presence of ST804 among clinical samples, which was predominant among resistant isolates between 1995–1999 and 2006–2010. Outside Brazil, an Australian study described its presence in sputum samples from CF patients [96]. Similarly, ST1602 was first detected in Rio de Janeiro in 2008, but retrospective data demonstrate its presence in clinical and environmental samples as early as 1995 [44,48]. STs 1859 and 1860 were identified in hospital effluent samples in 2010 by Miranda et al. [42]. Additionally, ST2236 and ST2237 were associated with invasive and cytotoxic virulotypes, respectively, among burn patients in a public hospital [43]. Lastly, the only non-HRC detected outside Rio de Janeiro, ST2524, was found to be clonally related to various types of infections among patients in adult, pediatric, and neonatal ICUs at a referral hospital in Pará State [49].

The present systematic review also identified several key mechanisms in *P. aeruginosa* that significantly contribute to the high levels of resistance phenotypes. The association of specific resistance genes and mutations with both HRCs and non-HRCs highlights the complexity of AMR in this pathogen. Except in the study by De Oliveira Luz et al. [51], all other studies include data on detection of β-lactamases, particularly as CR is further exacerbated by the acquisition of carbapenemases, including metallo-β-lactamases (MβLs) and serin-β-lactamases. For example, the *bla*_SPM-1_ gene, which encodes one such enzyme, has been widely reported across Brazilian studies as deeply investigated by our research group [33]. Other notable carbapenemases include *bla*_KPC_, *bla*_IMP_, and *bla*_VIM_, which have been detected at varying frequencies in different studies. For example, *bla*_KPC_ was found in 34.1% of isolates in the study by Miranda et al. [42] and 5.68% in the study by De Oliveira Santos et al. [48], reflecting the complex resistance profiles these isolates can exhibit. Even though not reported in the included studies, the *bla*_NDM_ and *bla*_KPC_ genes have been increasingly reported in Brazil since 2018, as a possible impact of COVID-19 [10,97]. Additionally, the overexpression of chromosomally encoded AmpC cephalosporinases has been identified as a significant resistance mechanism in studies by Cacci et al. [44] and Cavalcanti et al. [41], which showed that up to 30% of the isolates exhibited this overexpression.

Another primary mechanism contributing to resistance in *P. aeruginosa* is the overexpression of resistance-nodulation-division (RND) family efflux pumps, such as MexAB-OprM and MexXY-OprM. These efflux pumps play a crucial role in both intrinsic and acquired resistance, as noted in several studies [41,44,46,48]. These studies reported the overexpression of these pumps in a significant proportion of the isolates, emphasizing their contribution to the multidrug resistance observed in clinical settings in the Southeast and Northeast regions. Moreover, the loss or decreased expression of the porin protein OprD, which is essential for carbapenem uptake, has been extensively documented. For instance, mutations and deletions in the *oprD* gene were reported with some studies indicating that these mutations were present in all CR isolates [41,44,48]. Aminoglycoside resistance is also a significant concern and is frequently mediated by AMEs and 16S rRNA methylases. For instance, the *aac(6′)-Ib* gene, which modifies aminoglycosides and contributes to resistance, was identified in 50% of isolates by Silva et al. [41] and in 100% of isolates by Martins et al. [46]. The *rmtD* gene, which confers resistance to aminoglycosides by methylating the 16S rRNA, was identified in 100% of the tested isolates [46,52]. The complexity of resistance in *P. aeruginosa* is further demonstrated by the variety of other resistance mechanisms identified, such as *fosA*, *catB7*, *cmx*, and *sul1*, which have been detected in multiple studies.

Regarding virulence, the studies conducted on *P. aeruginosa* isolates from Brazil have highlighted the critical role of the T3SS in bacterial pathogenicity. The T3SS is a key virulence mechanism that enables *P. aeruginosa* to inject effector proteins directly into host cells, thereby modulating host cell processes and contributing to disease severity [15]. Among the six studies that specifically investigated virulence-related genes, a significant focus was placed on the detection of the *exoS*, *exoU*, *exoT*, and *exoY* genes, which encode T3SS effector proteins. The *exoS* gene encodes a bifunctional toxin that disrupts host cell signaling and contributes to cytotoxicity and immune evasion [98,99]. This gene was the most frequently detected T3SS-related gene among the studied isolates. In contrast, the *exoU* gene encodes a potent cytotoxin associated with more severe clinical outcomes, including rapid tissue damage and increased mortality rates [17,100]. The lower prevalence of *exoU* compared to *exoS* suggests variability in virulence strategies among different *P. aeruginosa* strains. Additionally, the *exoT* and *exoY* genes were identified in most of the isolates. *ExoT* has similar activities to *exoS*, affecting cytoskeleton integrity and promoting apoptosis, while *exoY* disrupts endothelial barriers and contributes to immune modulation [15,100,101]. Overall, these findings emphasize the need for the ongoing surveillance of virulence factors in *P. aeruginosa* to understand their distribution, impact on clinical outcomes, and potential role in guiding treatment decisions.

Despite these insights, significant heterogeneity was observed among the included studies, particularly from the Southeast region. This heterogeneity could be attributed to the diverse study designs, varying sample sizes, and methodological approaches employed across the studies. The meta-analysis results indicate that such variability is statistically significant, as reflected in the high I^2^ values for both HRC and ST277 analyses. The Southeast region’s data demonstrated a wide range of results, contributing to its significant heterogeneity (I^2^ = 87.11%). This variability might reflect regional differences in healthcare practices, antibiotic use, and environmental factors influencing the spread and prevalence of HRCs, particularly ST277. The higher prevalence rate of ST277 in this region could be indicative of localized outbreaks or adaptations, stressing the need for targeted public health interventions.

The Northern region exhibited the highest prevalence of HRCs, suggesting a unique epidemiological scenario possibly influenced by local environmental factors and hospital infection control practices. The substantial heterogeneity observed in the North (I^2^ = 89.37%) might also be a result of differing study designs and sampling periods. The presence of MDR clones is often associated with outbreaks in hospitals, where transmission between patients and environmental contamination can occur. In this sense, this finding in the Northern region of Brazil may be associated with the high rate of infections associated with this bacterium in healthcare institutions in the region [33,102]. Furthermore, environmental conditions, such as the contamination of surface waters due to its aquatic bacterial profile and the variation between rainy and dry periods may influence the presence of *P. aeruginosa* in waters of North regions [103]. Indeed, BIM-1, a novel acquired MBL belonging to subclass B1, was identified in an MDR *Pseudomonas* #2 strain recovered from a river in the state of Pará, demonstrating the possible spread of AMR markers in this context [104].

The Northeast region showed more uniformity in study findings, with low heterogeneity and a moderate prevalence of HRCs, likely due to smaller, more consistent study samples. Worryingly, a lack of data from the Central-West and South regions of Brazil was evidenced, which presents a significant limitation in this meta-analysis. The absence of studies from these regions means that the findings may not fully represent the national genetic landscape of *P. aeruginosa*. Future research should aim to address these gaps by conducting more recent and region-specific studies.

The studies included in this meta-analysis evaluated samples spanning from a wide range of years (1994–2021), and changes in molecular epidemiology and resistance patterns over time may not be fully captured, adding to the complexity of drawing uniform conclusions. The evolving nature of AMR necessitates continuous monitoring and updated research efforts. Overall, the meta-analysis highlights the importance of considering regional and temporal factors when assessing the molecular epidemiology of *P. aeruginosa*. The absence of publication bias suggests that the observed heterogeneity is due to true variations in study conditions and regional epidemiology rather than selective reporting. Future research should focus on recent data and standardized methodologies to better capture the evolving landscape of *P. aeruginosa* resistance and spread in Brazil.

## 5. Conclusions

The significance of HRCs such as ST235, ST244, ST111, ST357, and particularly ST277, along with non-HRCs, in the molecular epidemiology of *P. aeruginosa* in Brazil cannot be overstated. The persistence and spread of these strains across diverse regions and environments—from clinical settings to broader ecological reservoirs—highlight the critical need for ongoing surveillance and vigilant monitoring. This study’s findings reveal significant disparities in the detection of these strains, emphasizing the importance of comprehensive tracking efforts. Notably, several other non-HRCs were reported for the first time in Brazil, suggesting that these STs may have evolved locally and are potentially endemic to the region. This evidenced their contribution to the unique genetic landscape of *P. aeruginosa* in the country and reinforces the need for targeted public health strategies to manage their impact.

Finally, this systematic review and meta-analysis also highlight the critical issue of AMR in *P. aeruginosa* in Brazil and the need for targeted infection control measures, effective antimicrobial stewardship, and regional strategies tailored to local epidemiological patterns. Continued research and investment in novel therapeutic options are essential to combat the growing threat of AMR in this clinically important pathogen. The integration of detailed data from individual studies enhances our understanding of the epidemiological landscape and underscores the importance of addressing regional variations in AMR management.

## Figures and Tables

**Figure 1 antibiotics-13-00983-f001:**
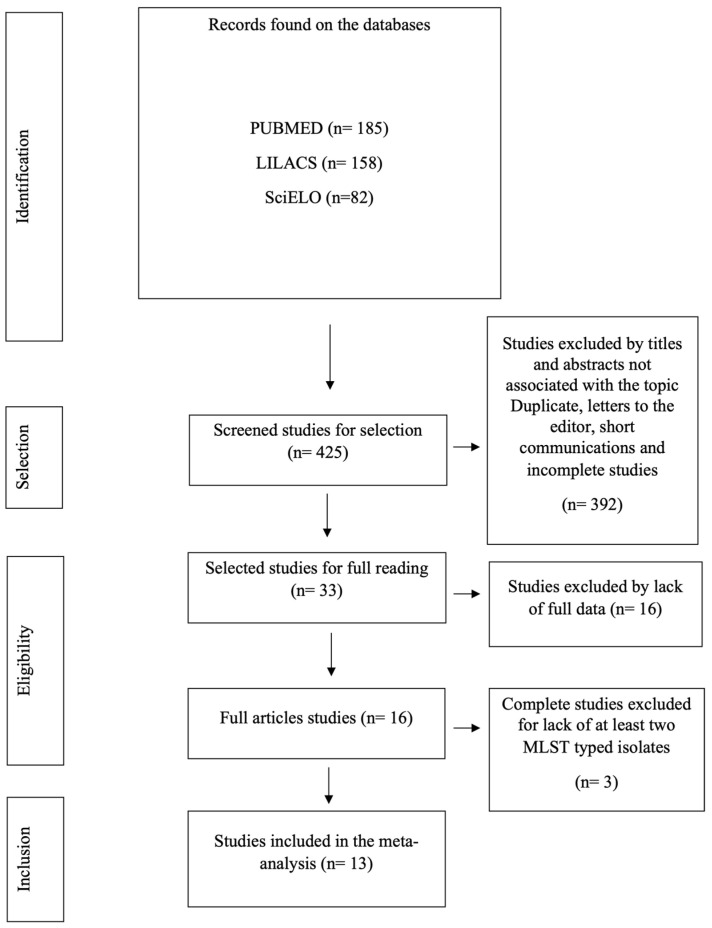
PRISMA flow diagram of the procedures for identifying, selecting, eligibility, and including studies for analysis. Belém, PA, Brazil, 2024.

**Figure 2 antibiotics-13-00983-f002:**
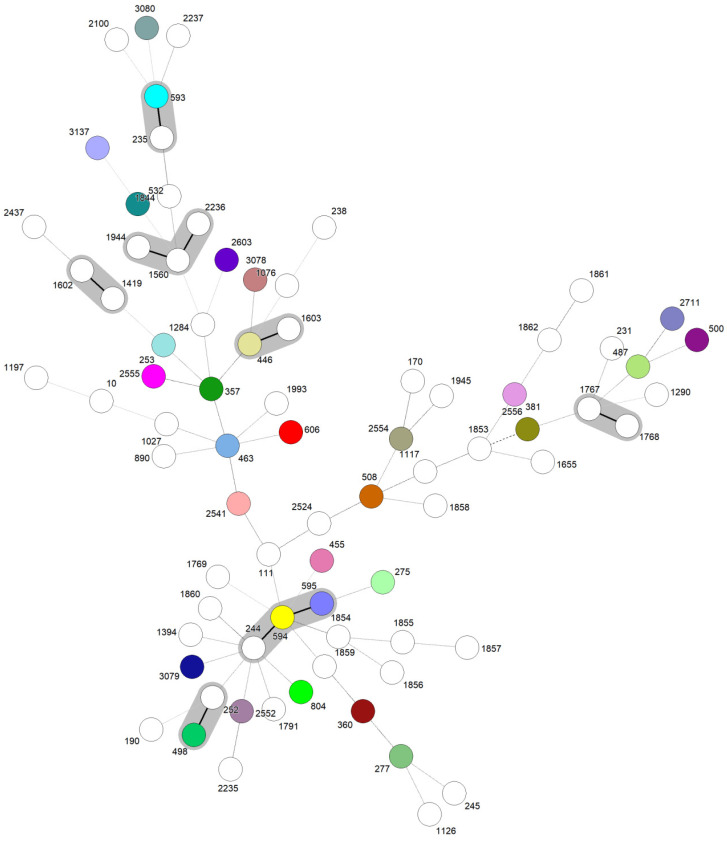
The goeBURST Full MST of *P. aeruginosa* demonstrating unique STs and clonal complexes in the Brazilian territory. Each node represents a unique ST; solid lines between STs shaded in grey represent SLVs and belong to CCs, and dashed lines between STs represent double locus variants DLVs. Colors were randomly assigned to STs.

**Figure 3 antibiotics-13-00983-f003:**
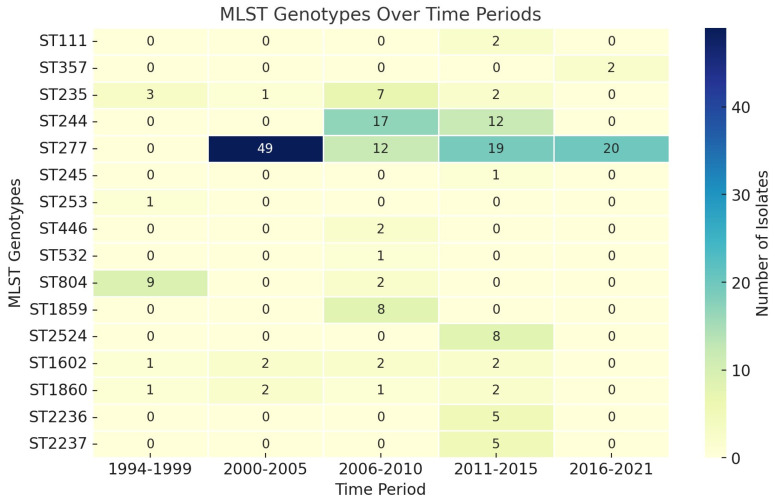
Distribution of the main *P. aeruginosa* MLST genotypes identified within studies across different time periods among isolates from 1994 to 2021.

**Figure 4 antibiotics-13-00983-f004:**
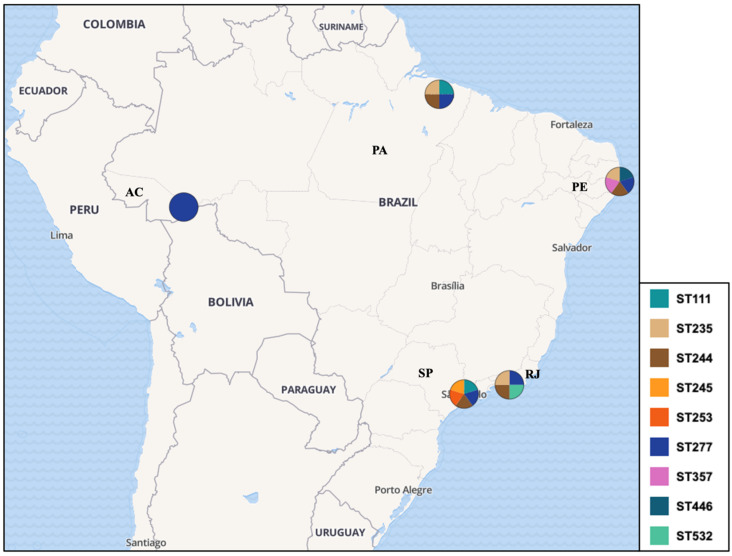
Distribution and regional spread of HRCs and emerging HRCs among *P. aeruginosa* isolates in Brazil. Legend: ST (sequence type), PA (Pará state), AC (Acre state), PE (Pernambuco state), SP (São Paulo state), RJ (Rio de Janeiro state).

**Figure 5 antibiotics-13-00983-f005:**
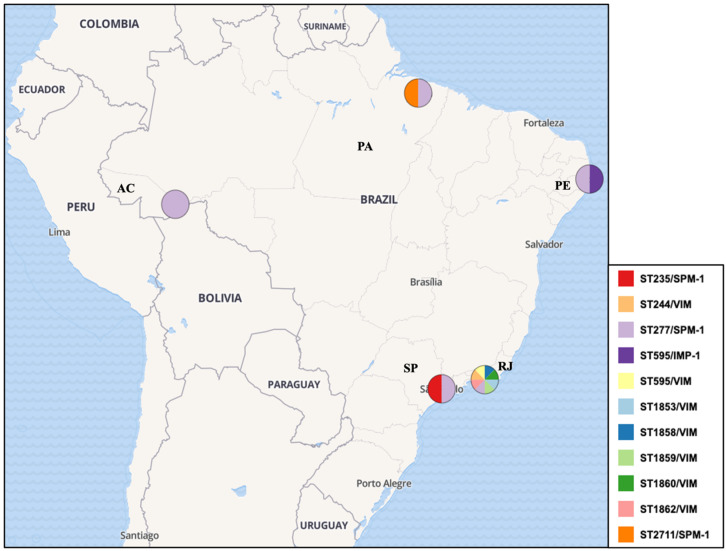
Distribution and regional spread of HRCs and non-HRCs of *P. aeruginosa* harboring MβL genes in Brazil. Legend: ST (sequence type), PA (Pará state), AC (Acre state), PE (Pernambuco state), SP (São Paulo state), RJ (Rio de Janeiro state).

**Figure 6 antibiotics-13-00983-f006:**
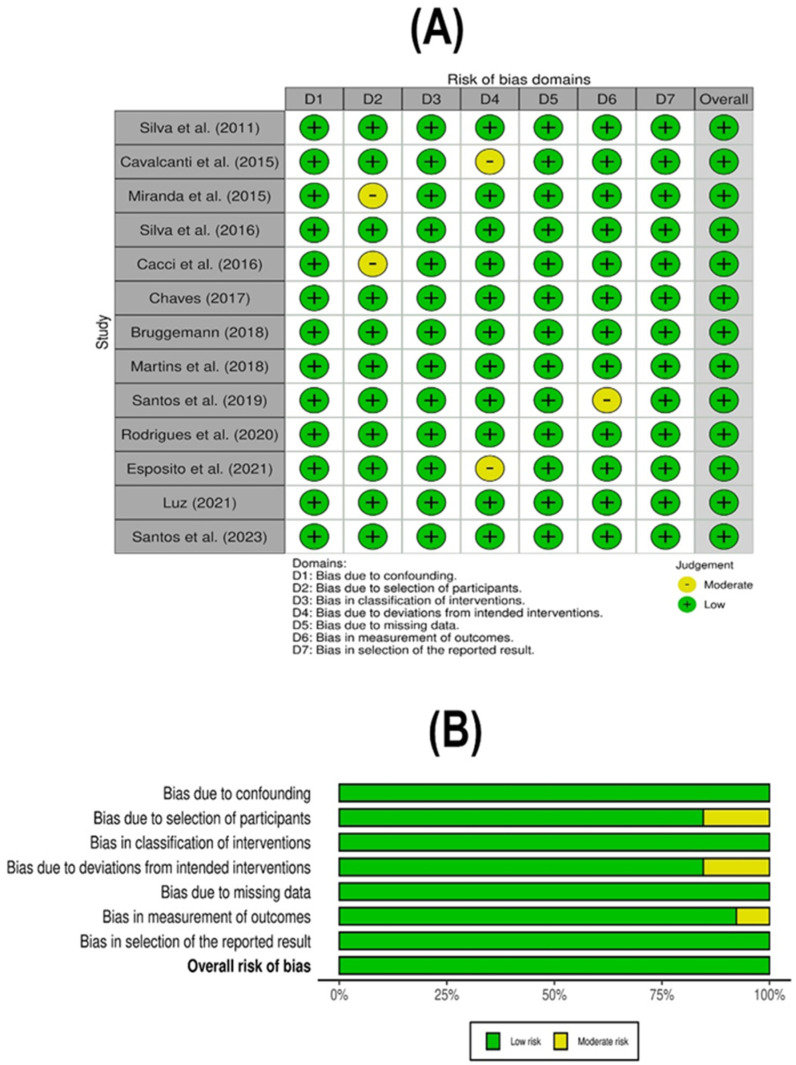
ROBINS-I Risk of Bias Assessment. Risk of bias assessment using the ROBINS-I tool for the observational studies included in the meta-analysis. (**A**) Evaluation of individual study parameters, and (**B**) combined data evaluation for each methodological parameter assessed [40,41,42,43,44,45,46,47,48,49,50,51,52].

**Figure 7 antibiotics-13-00983-f007:**
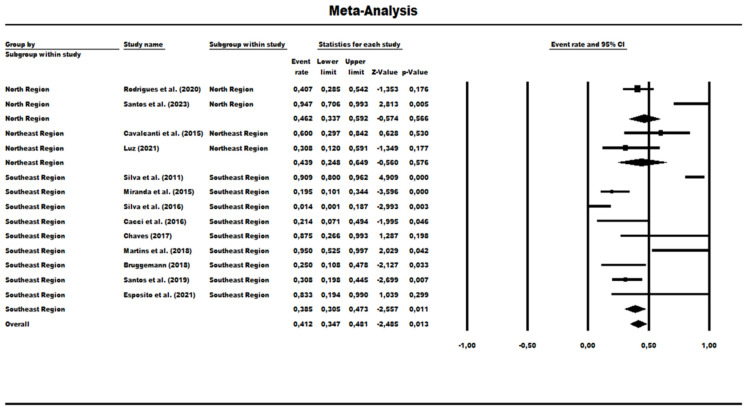
Forest plot of HRCs’ prevalence among *P. aeruginosa* isolates in Brazil and geographical regions [40,41,42,43,44,45,46,47,48,49,50,51,52].

**Figure 8 antibiotics-13-00983-f008:**
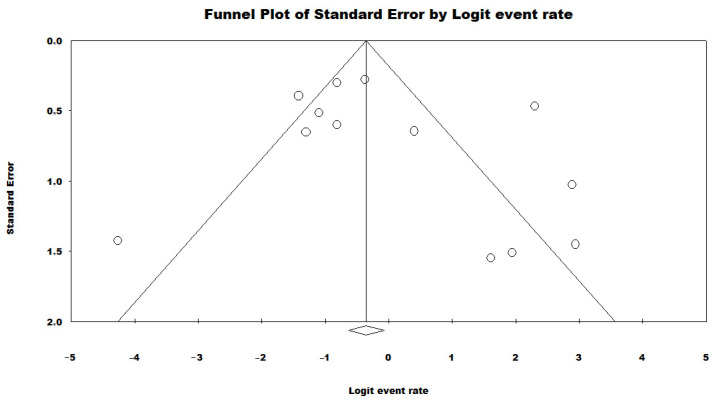
Funnel plot evaluating potential publication bias in the meta-analysis of the proportion of HRCs and non-HRCs among *P. aeruginosa* isolates in Brazil [40,41,42,43,44,45,46,47,48,49,50,51,52]. Each circle typically represents an individual study included in the meta-analysis. The position of the circle on the plot indicates the estimated effect size of that study (on the x-axis) and its precision (on the y-axis), often measured by standard error or sample size. Rhombuses may be used to highlight certain studies within the funnel plot. For instance, they can represent studies that are considered outliers or those that have been flagged for having methodological issues or other biases.

**Figure 9 antibiotics-13-00983-f009:**
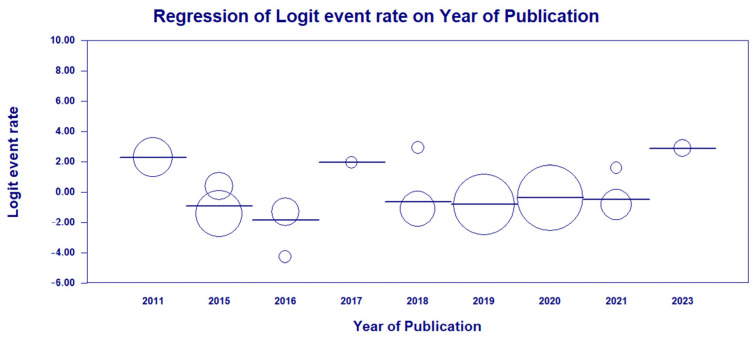
Meta-regression analysis examining the relationship between the year of publication and HRCs and non-HRCs of *P. aeruginosa* based on the studies included in the meta-analysis [40,41,42,43,44,45,46,47,48,49,50,51,52].

**Figure 10 antibiotics-13-00983-f010:**
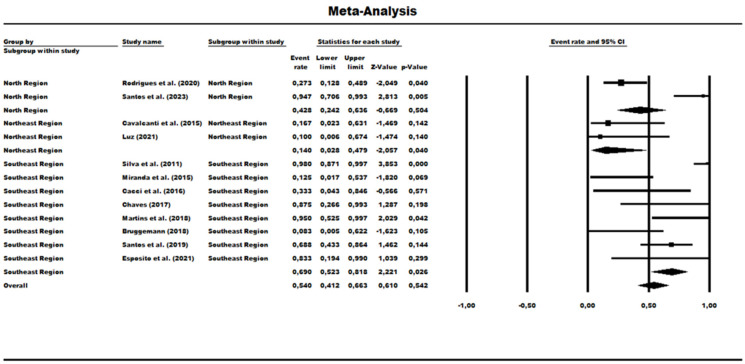
Forest plot displaying the prevalence of the ST277 genotype among HRCs of *P. aeruginosa* isolates in Brazil [40,41,42,44,45,46,47,48,49,50,51,52].

**Figure 11 antibiotics-13-00983-f011:**
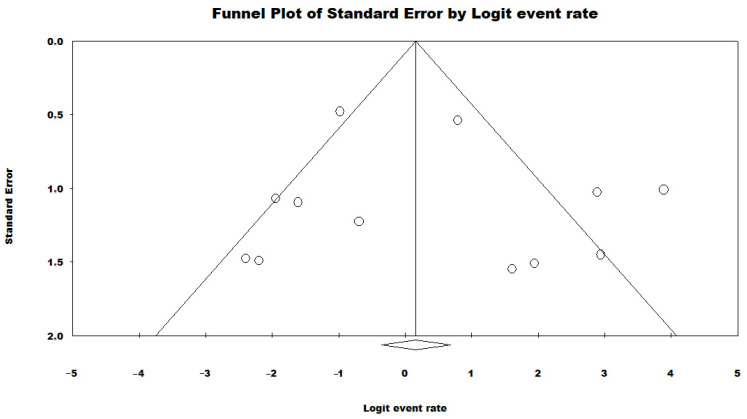
Funnel plot assessing potential publication bias in the meta-analysis of the prevalence of the ST277 genotype among HRCs of *P. aeruginosa* in Brazil [40,41,42,44,45,46,47,48,49,50,51,52]. Each circle typically represents an individual study included in the meta-analysis. The position of the circle on the plot indicates the estimated effect size of that study (on the x-axis) and its precision (on the y-axis), often measured by standard error or sample size. Rhombuses may be used to highlight certain studies within the funnel plot. For instance, they can represent studies that are considered outliers or those that have been flagged for having methodological issues or other biases.

**Figure 12 antibiotics-13-00983-f012:**
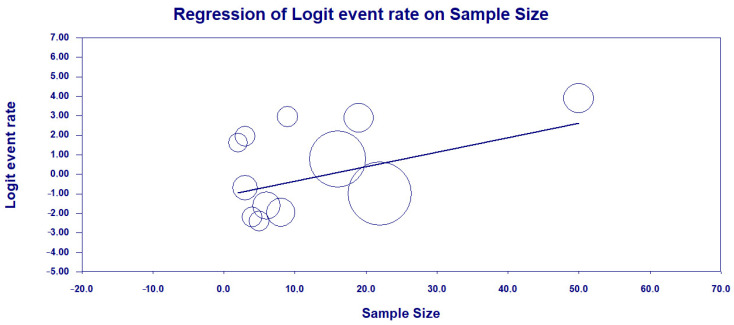
Meta-regression analysis examining the influence of sample size on the prevalence of the ST277 genotype among HRCs of *P. aeruginosa* based on the studies included in the meta-analysis [40,41,42,44,45,46,47,48,49,50,51,52].

**Table 1 antibiotics-13-00983-t001:** Characteristics of the included studies in this present meta-analysis.

Authors/Year	Type of Study	Isolates Period	Region/Location	Setting	N° of Samples (872)/ N° of MLST Typed (298)	MLST Genotypes	HRC	JBI Score
Silva et al. (2011) ^#,^* [40]	Retrospective cross-sectional	2000–2004	Southeast region	Clinical	55/55	ST277 (49/55—89.09%) ST235 (1/55—1.82%) ST593 (3/55—5.45%) ST594 (1/55—1.82%) ST595 (1/55—1.82%)	Yes—50/55 (90.9%)—ST277, ST235	(7/8)
Cavalcanti et al. (2015) ^#,^* [41]	Retrospective cross-sectional	2008–2010	Northeast region	Clinical	10/10	ST235 (3/10—30.0%) ST446 (1/10—10.0%) ST1560 (1/10—10.0%) ST1419 (1/10—10.0%) ST1126 (1/10—10.0%) ST277 (1/10—10.0%) ST244 (2/10—20.0%)	Yes—6/10 (60.0%)—ST277, ST235, ST244	(8/8)
Miranda et al. (2015) ^#,^* [42]	Retrospective cross-sectional	2008 and 2010	Southeast region	Hospital wastewater treatment plant	41/36	ST244 (8/36—22.22%) ST1853 (2/36—5.56%) ST1854 (1/36—2.78%) ST1855 (2/36—5.56%) ST1856 (1/36—2.78%) ST1857 (1/36—2.78%) ST1858 (2/36—5.56%) ST1859 (8/36—22.22%) ST1860 (1/36—2.78%) ST1861 (1/36—2.78%) ST1862 (4/36—11.11%) ST238 (1/36—2.78%) ST381 (1/36—2.78%) ST595 (3/36—8.33%)	Yes—8/41 (19.51%)—ST244	(6/8)
Silva et al. (2016) ^#,^* [43]	Retrospective cross-sectional	2012	Southeast region	Burn center: Clinical and Clinical environment	35/10	ST 2236 (5/10—14.28%) ST 2237 (5/10—14.28%)	No (0.0%)	(8/8)
Cacci et al. (2016) ^#,^* [44]	Retrospective cross-sectional	2007–2008	Southeast region	Clinical and Clinical environment	482/14	ST1027 (1/14—7.14%) ST1602 (2/14—14.29%) ST1603 (1/14—7.14%) ST1767 (1/14—7.14%) ST1768 (1/14—7.14%) ST1769 (1/14—7.14%) ST1844 (1/14—7.14%) ST235 (1/14—7.14%) ST244 (1/14—7.14%) ST277 (1/14—7.14%) ST446 (1/14—7.14%) ST532 (1/14—7.14%) ST890 (1/14—7.14%)	Yes—(3/14—21.42%)—ST235, ST244 and ST277	(8/8)
Chaves et al. (2017) ^#,^* [45]	Retrospective case-control study	2011–2013	Southeast region	Clinical	29/3	ST277 (3/3—100.0%)	Yes—ST277 (3/3—100.0%)	(10/10)
Martins et al. (2018) ^#,^* [46]	Retrospective cross-sectional	2012	Southeast region	Environmental (Migratory birds)	9/9	ST277: 9/9 (100%)	Yes—9/9 (100.0%)—ST277	(7/8)
Bruggemann et al. (2018) ^#,^* [47]	Retrospective cross-sectional	1994–2016	Southeast region	Clinical	20/20	ST235 (3/20—15.0%) ST10 (1/20—5.0%) ST231 (1/20—5.0%) ST245 1/20—5.0%) ST252 (1/20—5.0%) ST446 (1/20—5.0%) ST455 (1/20—5.0%) ST498 (1/20—5.0%) ST606 (1/20—5.0%) ST1290 (1/20—5.0%) ST1993 (1/20—5.0%) ST2235 (1/20—5.0%) ST381 (1/20—5.0%) ST190 (1/20—5.0%) ST244 (2/20—10.0%)	Yes—5/20 (25.0%)—ST235, ST244	(8/8)
Santos et al. (2019) ^#,^* [48]	Retrospective cross-sectional	1995–2015	Southeast region	Clinical	88/52	ST1117 (3/52—5.77%) ST1560 (3/52—5.77%) ST1945 (3/52—5.77%) ST244 (5/52—9.62%) ST277 (11/52—21.15%) ST487 (4/52—7.69%) ST1791 (3/52—5.77%) ST1860 (4/52—7.69%) ST804 (11/52—21.15%) ST1602 (3/52—5.77%) ST1944 (2/52—3.85%)	Yes—16/52 (30.76%)—ST277, ST244	(7/8)
Rodrigues et al. (2020) ^#,^* [49]	Retrospective cross-sectional	2010–2013	North region	Referral hospital/Clinical	54/54	ST2524 (8/54—14.81%) ST2541 (2/54—3.70%) ST2552 (2/54—3.70%) ST2554 (1/54—1.85%) ST2555 (1/54—1.85%) ST2556 (2/54—3.70%) ST2603 (1/54—1.85%) ST111 (2/54—3.70%) ST170 (1/54—1.85%) ST235 (5/54—9.26%) ST244 (9/54—16.67%) ST277 (6/54—11.11%) ST360 (1/54—1.85%) ST463 (1/54—1.85%) ST500 (2/54—3.70%) ST508 (1/54—1.85%) ST1076 (1/54—1.85%) ST1197 (1/54—1.85%) ST1284 (4/54—7.41%) ST1655 (1/54—1.85%) ST2100 (1/54 —1.85%) ST2437 (1/54—1.85%)	Yes—22/54 (40.7%)—ST244, ST277, ST111, ST235	(8/8)
Esposito et al. (2021) ^#^ [50]	Retrospective cross-sectional	2016	Southeast region	Impacted urban rivers (environmental)	2/2	ST277 (2/2—100.0%)	Yes—(2/2—100.0%) ST277	(8/8)
De Oliveira Luz et al. (2021) ^#,^* [51]	Retrospective cross-sectional	2010, 2015, and 2016	Northeast region	Clinical	13/13	ST244 (2/13—15.38%) ST1394 (1/13—7.69%) ST252 (1/13—7.69%) ST3079 (3/13—23.07%) ST357 (2/13—15.38%) ST3078 (1/13—7.69%) ST275 (1/13—7.69%) ST3080 (1/13—7.69%) ST3137 (1/13—7.69%)	Yes—(4/13—30.76%)—ST244, ST357	(8/8)
Santos et al. (2023) ^#^ [52]	Retrospective cross-sectional	2018–2021	North region	Clinical	34/19	ST277 (18/19—94.74%) ST2711 (1/19—5.26%)	Yes—(18/19—94.74%)—ST277	(8/8)

Legend: * Retrieved from PubMed database based on search strategy; ^#^ Retrieved from LILACS database based on search strategy; ST (sequence type).

## Data Availability

The original contributions presented in the study are included in the article/Appendix A, further inquiries can be directed to the corresponding authors.

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
