# Peer review of "Molecular Epidemiology of Pseudomonas aeruginosa in Brazil: A Systematic Review and Meta-Analysis"

_antibiotics, 2024, doi:10.3390/antibiotics13100983_

Round 1
Reviewer 1 Report
Comments and Suggestions for Authors
The meta-analysis carried out by the authors complies with the established methodology, and the rigor of their criteria determines the veracity of the results.
Just a few suggestions for the authors
Instead of using Phylogenetic tree based on the Neighbor-Joining algorithm, I suggest Phylogenetic network because of the high recombination of Pseduomonas aeruginosa during infectious processes and that many of the isolates from the same location are clonally related. The story of the high-risk clones in the identified environment and how they adapt to the environments could be better told.
Table 2, which describes the characteristics of the 872 strains.
It is not clear. It would be worth putting the complete data by strain in the supplementary material, if all strains are used, but when adding the number of strains by variable, they never add up to 872. Please review, correct and, failing that, complete a table with the strains and variants of the study. Perhaps what is relevant are the 298 strains that have ST.
Author Response
Dear Reviewer,
Thank you for your thorough evaluation and insightful suggestions. We have carefully considered your feedback and made revisions accordingly. Below is our point-by-point response to your comments.
- Instead of using Phylogenetic tree based on the Neighbor-Joining algorithm, I suggest Phylogenetic network because of the high recombination of Pseduomonas aeruginosa during infectious processes and that many of the isolates from the same location are clonally related.
Response: We appreciate your suggestion, and after consideration, we have decided to exclude Figure 3 presenting the phylogenetic tree based on the Neighbor-Joining algorithm and maintain the phylogenetic network based on Minimum Spanning Tree (MST) approach on Figure 2. The MST effectively represents the clonal relationships and genetic distance between isolates, and given the focus of our study, the MST provides a clearer and simpler visualization of the data.
- The story of the high-risk clones in the identified environment and how they adapt to the environments could be better told.
Response: Thank you for your insightful comment. Upon careful review of the manuscript, we believe that the mechanisms of adaptation of high-risk clones (Pseudomonas aeruginosa) to their environments have already been well presented. The current text provides detailed explanations of how these clones adapt through biofilm formation, acquisition of resistance genes via mobile genetic elements, and other mechanisms such as efflux pump overexpression and porin mutations. Furthermore, the manuscript discusses the presence of these clones in both clinical and non-clinical environments, such as impacted urban rivers and hospital wastewater systems, which illustrates their ability to thrive in diverse ecological niches. Given this, we respectfully suggest that no further elaboration is necessary
- Table 2, which describes the characteristics of the 872 strains. It is not clear. It would be worth putting the complete data by strain in the supplementary material, if all strains are used, but when adding the number of strains by variable, they never add up to 872. Please review, correct and, failing that, complete a table with the strains and variants of the study. Perhaps what is relevant are the 298 strains that have ST.
Response: After careful consideration, we have decided to exclude the Table 2 from the main text. Instead, we presented all available data to the studies to the supplementary material.
Reviewer 2 Report
Comments and Suggestions for Authors
The study by Rodrigues et al. is presented as a systematic review and meta-analysis on the molecular epidemiology of Pseudomonas aeruginosa, with the main objective of determining the distribution of high-risk and non-high-risk clones in Brazil. The discussion is broad and detailed, highlighting the most important aspects of the identified sequence types and their epidemiological relevance, their relationship with the dissemination of antimicrobial resistance (AMR), and the pathogenicity of P. aeruginosa. Perhaps the discussion is too extensive and could be condensed to emphasize the most relevant points.
The reported results have local relevance, as the most significant data on HRC have already been systematized in previous publications, some of them very recent.
Regarding the methodology, I would like clarification on why the terms "Humans" OR "Animals" were included in the search. This may have restricted the studies retrieved.
Specific Comments:
Line 19: several intrinsic and ACQUIRED antimicrobial resistance
Line 56: the third (3rd) MOST isolated bacterial pathogen.
Choose between "third" and "3rd"; it is unnecessary to include both expressions.
Lines 61 to 64: Review the phrasing; it is confusing.
Line 65: mechanisms OF P. aeruginosa
Line 96: Change "in the Brazilian area" to "in Brazil."
Line 237: aminoglycoside-modifying enzymes (AMEs) (without an apostrophe). Also correct this in Table 2.
Line 361: This statement is debatable, as the search was restricted with the terms "Humans" OR "Animals."
Line 377: Correct the nomenclature of beta-lactamases; the enzyme name should be in subscript.
Line 385: Indicate the surname of the author of the cited publication [52].
Line 491: If the class of the CRISPR/Cas system is not indicated, use "CRISPR/Cas systems" to refer to CRISPR/Cas systems in general.
Line 500: Note that blaPAO and bla50 and their derivatives are not horizontally acquired beta-lactamase genes.
Line 536: SERIN-beta-lactamases
Author Response
Dear Reviewer,
Thank you for your insightful evaluation and suggestions. We have carefully considered your feedback and made revisions accordingly. Below is our point-by-point response to your comments.
- The study by Rodrigues et al. is presented as a systematic review and meta-analysis on the molecular epidemiology of Pseudomonas aeruginosa, with the main objective of determining the distribution of high-risk and non-high-risk clones in Brazil. The discussion is broad and detailed, highlighting the most important aspects of the identified sequence types and their epidemiological relevance, their relationship with the dissemination of antimicrobial resistance (AMR), and the pathogenicity of aeruginosa. The reported results have local relevance, as the most significant data on HRC have already been systematized in previous publications, some of them very recent. Perhaps the discussion is too extensive and could be condensed to emphasize the most relevant points.
Response: We have reviewed and condensed as much as possible the discussion as requested.
- Regarding the methodology, I would like clarification on why the terms "Humans" OR "Animals" were included in the search. This may have restricted the studies retrieved.
Response: Searching with the OR operator does not restrict a systematic search because, instead of limiting the results to documents that contain all the specified terms, it allows the inclusion of any of the terms. This results in a broadening of the search, increasing the number of references retrieved.
- Specific Comments:
Line 19: "acquire" has been corrected to "acquired."
Line 56: We now use "third" instead of "third (3rd)" for consistency.
Lines 61 to 64: Corrected as requested.
Line 65: Corrected as requested.
Line 96: Changed "in the Brazilian area" to "in Brazil."
Line 237: The term "aminoglycoside-modifying enzymes (AMEs)" has been corrected.
Line 361: The statement is correct as the search as not restricted.
Line 377: Beta-lactamase nomenclature is now presented in subscript and reviewed in the whole manuscript.
Line 385: The cited author’s surname has been added.
Line 491: We now refer to "CRISPR/Cas systems" to cover the general case.
Line 536: Corrected to "Serine-beta-lactamases."
Reviewer 3 Report
Comments and Suggestions for Authors
Authors should correct the following items:
- In the abstract, line 30 the abbreviation of "STs" has not been written out in full;
- In the introduction, lines 82 to 84, the sentence should be reworded because it does not reflect the correct meaning. The reader is left with the idea that the MLST technique was developed by Curran and colleagues, when what the authors wanted to indicate is that the application of the MLST technique to the genotyping of P. aeruginosa was developed by Curran and colleagues.
The following typographical errors should be corrected:
- In the caption of figure 1 "Flow and Diagram" should be in lowercase letters;
- In the caption of figure 2 “Full” should be in lowercase letters and remove the bold from the caption
- In the captions of figures 5 and 6 remove the bold from the caption
- In the caption of figure 7, use lowercase letters without bold.
- In the caption of figure 8 all words except the first one and “HRCs” must begin with a lowercase letter
- In the discussion in line 507 correct "has be" to "has been"
- In the discussion, line 562 in "the tested isolates tested" delete the second "tested"
Comments on the Quality of English LanguageThe English language is adequate and understandable, with small typographical errors that i identify to the authors for correction
Author Response
Dear Reviewer,
Thank you for your insightful evaluation and suggestions. We have carefully considered your feedback and made revisions accordingly. Below is our point-by-point response to your comments.
Authors should correct the following items:
- In the abstract, line 30 the abbreviation of "STs" has not been written out in full;
Response: Corrected as requested.
- In the introduction, lines 82 to 84, the sentence should be reworded because it does not reflect the correct meaning. The reader is left with the idea that the MLST technique was developed by Curran and colleagues, when what the authors wanted to indicate is that the application of the MLST technique to the genotyping of aeruginosawas developed by Curran and colleagues.
Response: Corrected as requested.
The following typographical errors should be corrected:
- In the caption of figure 1 "Flow and Diagram" should be in lowercase letters;
Response: Corrected as requested.
- In the caption of figure 2 “Full” should be in lowercase letters and remove the bold from the caption
Response: Corrected as requested.
- In the captions of figures 5 and 6 remove the bold from the caption
Response: Corrected as requested.
- In the caption of figure 7, use lowercase letters without bold.
Response: Corrected as requested.
- In the caption of figure 8 all words except the first one and “HRCs” must begin with a lowercase letter
Response: Corrected as requested.
- In the discussion in line 507 correct "has be" to "has been"
Response: Corrected as requested.
- In the discussion, line 562 in "the tested isolates tested" delete the second "tested"
Response: Corrected as requested.